# Wind turbine load dynamics in the context of turbulence intermittency

Carl Michael Schwarz[1], Sebastian Ehrich[1], and Joachim Peinke[1,2]

[1]ForWind, Institute of Physics, Carl-von-Ossietzky University Oldenburg, Küpkersweg 70, 26129 Oldenburg, Germany
[2]Fraunhofer Institute for Wind Energy Systems, Küpkersweg 70, 26129 Oldenburg, Germany

**Correspondence:** C. M. Schwarz (carl.michael.schwarz@uol.de)

**Abstract.** The importance of a high order statistical feature of wind, which is neglected in common wind models, is investigated: Non-Gaussian distributed wind velocity increments related to the intermittency of turbulence and their impact on wind turbines dynamics and fatigue loads are in the focus. Gaussian and non-Gaussian synthetic wind fields obtained from a Continuous-Time-Random Walk model are compared and fed to a common aero-servo-elastic model of a wind turbine employing Blade-Element/Momentum (BEM) aerodynamics. It is discussed why and how the effect of the non-Gaussian increment statistics has to be isolated. This is achieved by assuring that both types feature equivalent probability density functions, spectral properties and coherence, which makes them indistinguishable based on wind characterizations of common design guidelines. Due to limitations in the wind field genesis idealized spatial correlations are considered. Three examples with idealized, differently sized wind structures are presented. A comparison between the resulting wind turbine loads is made. For the largest wind structure sizes differences in the fatigue loads between intermittent and Gaussian are observed. These are potentially relevant in a wind turbine certification context. Subsequently, the dependency of this intermittency effect on the field's spatial variation is discussed. Towards very small structured fields the effect vanishes.

## 1 Introduction

Exact representations of wind and its dynamics are essential to the planning and design of wind turbines. So called *wind models* are utilized to describe the dynamic behaviour. Guidelines, such as the International Electrotechnical Commission (IEC) standard 61400-1 for wind turbines (International Electrotechnical Commision), feature for example the Kaimal (Kaimal et al., 1972) and Mann (Mann, 1994) models. In addition to these models for the wind fluctuations, deterministic wind gusts are considered to account for extreme events, such as the so called *50 years gust*. The need for these additional wind scenarios indicates the incompleteness of wind models: Due to the high complexity of wind, modelling efforts are often focusing on specific features, while neglecting others, as they are assumed to be of minor importance. The aforementioned examples focus on representing the spectral properties of wind velocity components and their coherence. This approach is widely accepted and reproduces the targeted features of wind well.

A known feature of wind which is not considered in common models is related to the statistics of wind speed increments $u_\tau(t)$. Mathematically $u_\tau(t)$ can easily be obtained from a given lag value $\tau$ and a wind time series $u(t)$ as

$$u_\tau(t) := u(t+\tau) - u(t). \tag{1}$$

A comprehensive introduction into the characterization of wind by these statistics has been given by Morales et al. (Morales et al., 2012). In short, the increments $u_\tau(t)$ can be understood intuitively as changes in wind speed and therefore as a simple indicator for gusts. In conclusion their dynamics might be relevant in the context of wind turbine performances and loads.

For wind it is well documented that $u_\tau(t)$ is non-Gaussian behaved (Böttcher et al., 2003; Vindel et al., 2008; Morales et al., 2012; Mücke et al., 2011). In this work this non-Gaussianity is referred to as the *intermittency of turbulence*[1]. However wind models are commonly based on spectral properties and imply Gaussian behaved $u_\tau(t)$, cf. e.g. (Powell and Connell, 1986; Mücke et al., 2011). As elaborated by Böttcher et al. (Böttcher et al., 2003) the assumption of Gaussian $u_\tau(t)$ leads to false predictions, especially about extreme events: Events predicted to occur only each 500 years by a Gaussian model, are predicted to occur five times a day if non-Gaussianity is considered. Thus the question arises, whether or not the omission of intermittency in the wind modelling process is a safe simplification. If intermittency would have a relevant impact on wind turbine loads this concept should be considered for implementation into future wind models. This work aims to answer this question by comparing intermittent wind fields against purely Gaussian ones with respect to their influence on wind turbine dynamics and loads.

This paper is organized as follows: Other studies have been dedicated to this issue. They are discussed in Sec. 1.1. The specific contribution of this work is described in Sec. 1.2. Detailed aspects about the approach of this work, especially about the wind modelling, are given in Sec. 2. The results are presented and discussed in Sec. 3. Summarizing conclusions are given in Sec. 4.

## 1.1 Literature overview

The discussion of non-Gaussian increments in the context of wind energy has been started by Böttcher et al. (Böttcher et al., 2001, 2003). Intermittency has also been observed in data obtained from field measurements (Böttcher et al., 2003; Morales et al., 2012; Vindel et al., 2008). Further efforts focusing on non-Gaussian power increment statistics of single wind turbines and entire wind parks have been made(Wächter et al., 2012; Milan et al., 2013; Hähne et al., 2018), which can be interpreted as a footprint of the statistics of interest. Milan et al. (Milan et al., 2013) were able to fit a non-linear scaling model including the intermittent wind dynamics to the scaling behaviour of both the wind and power output data, relating both of these dynamics. Lately, Hähne et al. (Hähne et al., 2018) reported on the intermittent power statistics of wind energy. These studies can be seen as evidence that intermittent dynamics of the wind are present in entire wind energy conversion process. A deeper insight into how the intermittent wind dynamics excite the turbine is not presented in these works.

---

[1]Other definitions of intermittency in the context of turbulence may be used.

In order to generate intermittent wind dynamics, a wind model has been developed by Kleinhans (Kleinhans, 2008). It relies on the concept of Continuous-Time Random Walks (CTRW) introduced by Montroll and Weiss (Montroll and Weiss, 1965), and has been applied in related studies presented in the following. It is further utilized in this work. Details are given in Section 2.

Pioneering work with respect to intermittency and wind turbine loads was conducted by Gontier et al. (Gontier et al., 2007). The authors tested two standard wind models (Kaimal (Kaimal et al., 1972), Mann (Mann, 1994)) and the intermittent CTRW wind model (Kleinhans, 2008) (cf. Section 2.1.1) with respect to their impact on fatigue loading of different sensors. Blade-Element-Momentum (BEM) theory based wind turbine computations were conducted. The authors drew conclusions to relevant load sensors, such as e.g. blade root bending moments and the tilt moment at the tower top. Differences in the fatigue

loads for the different models were detected and described, but could not be embedded into a clear overall trend. Although the direct comparison of different wind models is interesting, these models feature fundamental differences that will affect the wind turbine loads, e.g. spectral properties. In other words: Intermittency is not isolated as the main difference between these wind fields. This represents an obstacle in drawing further conclusions from the presented study, as the reason for the deviations in the results obtained for different wind fields could also be a consequence of other statistical differences.

Mücke et al. (Mücke et al., 2011) adopted aspects of the methodology applied by Gontier et al. and added a comparison against measured wind data. Three types of wind field data were used: A measured data set from the GROWIAN site (Körber et al., 1988), a common Kaimal model (Kaimal et al., 1972; International Electrotechnical Commision) and the CTRW model (Kleinhans, 2008). All types of wind fields are processed with a purely aerodynamic BEM based wind turbine model. The CTRW fields were designed in order to have similar increment statistics as the GROWIAN fields. For all types of fields a

high correlation between the wind increments statistics and the resulting torque increment statistics was reported. The authors showed that non-Gaussian wind statistics can lead to non-Gaussian torque statistics. A Rain-Flow-Counting (RFC) analysis (Matsuishi and Endo, 1968) was conducted on the resulting torque data of the GROWIAN measurement field and the Kaimal field. It is concluded that the RFC method is not sensitive to the intermittent dynamics, as a certain amount of temporal information is lost within a RFC procedure. However revisiting the results by Mücke et al., differences in the load range histogram

especially at higher load ranges are evident which, when potentiated with the SN-slope coefficient[2], could lead to pronounced differences in the Equivalent Fatigue Loads (EFL). Therefore the conclusion that common fatigue estimation procedures, such as the RFC, are insensitive to intermittency is arguable. Mücke et al. add the comparison against measured wind data to the discussion, which is of high interest, but also challenging: Measured wind data commonly contains instationarities and trends, while wind models usually yield stationary time series.

A different approach to obtain intermittent wind fields was utilized in the study presented by Berg et al. (Berg et al., 2016). The authors investigate wind fields derived from Large-Eddy-Simulations (LES) of the Atmospheric Boundary Layer (ABL). Snapshots of 'frozen', three-dimensional velocity fields, exhibiting the intermittent dynamics, were extracted from the simulation result. The three spatial dimensions are converted into an unsteady, two dimensional velocity plane via Taylor's Hypothesis of frozen turbulence and processed through a common aero-servo-elastic model of a wind turbine. Gaussian fields were ob-

---

[2]See Section 2.3 for further information.

tained by deriving surrogate fields based on Proper Orthogonal Decomposition (POD) of the original data. In doing so, the exact same second-order statistics were obtained for the surrogate, non-intermittent fields. 20 fields of each type were processed through a BEM based aero-elastic wind turbine model in order to evaluate the impact of intermittency on wind turbine loads. Both ultimate and fatigue loads resulting from these simulations were compared. Based on an RFC-based fatigue analy-
sis and an analysis of global load extrema the authors do not find any significant evidence that intermittency alters the loads and therefore conclude that the relevant dynamics are low-pass filtered by the turbine, as they are mainly found in small structures below the rotor scale. The work by Berg et al. successfully delivers an approach that respects other statistics of wind fields and aims at the isolation of intermittency: The approach of generating a pair of wind fields with the same statistical properties – aside from the distribution of two-point statistics – is the preferable approach to analyse the impact of intermittency on wind
turbine loads. This work follows this approach.

Schottler et al. (Schottler et al., 2017) compared Gaussian and non-Gaussian wind fields in an experimental campaign featuring a model wind turbine. The turbulent fluctuations in the experiment were achieved with an active grid. The authors compare the response of the model wind turbine to two different kinds of inflows: One with Gaussian, the other with non-Gaussian increment statistics. Both inflows are equivalent with respect to their standard deviation and mean value. The authors demon-
strate that the turbine's response (e.g. the rotor thrust) still contains the non-Gaussian dynamics, contradicting the conclusion by Berg et al. (Berg et al., 2016) that these dynamics are filtered by the turbine. However, Schottler et al. do not report on the size of the wind structures in the tested wind fields, which is related to the argumentation Berg et al. and is also in the focus of this work. Secondly only a few statistical wind parameters could be controlled and made comparable, which hamper the isolation of intermittency (cf. Sec.2.1.2) and the ability to draw conclusions with respect to intermittency.

In a recent national project (Wächter et al., 2017) the effect of intermittent wind dynamics on fatigue loads was tested. Synthetic Kaimal (Kaimal et al., 1972), Mann (Mann, 1994) and intermittent CTRW Fields (Kleinhans, 2008) were processed in a BEM-based wind turbine model. Subsequently, the resulting load time series were used as an excitation signal in an experimental set-up, in which material probes were dynamically loaded until failure. Significant differences were observed, indicating a much faster damage accumulation for intermittent wind. However due to differences in the fields with respect to their spectral
properties and coherence, these findings cannot be attributed to the non-Gaussian increment statistics exclusively. The application of intermittent dynamics to material probes however is highly interesting, as one does not rely on fatigue estimation procedures, as the sensitivity of these procedures to intermittent dynamics is at question.

In summary, contradicting conclusions with respect to the effect of intermittency on wind turbines have been drawn. It is
still unclear how to judge the impact of these statistics, as some questions are still left open. Also, there is a need to discuss the proper isolation of intermittency within this issue.

## 1.2 Contribution of this work

This work aims to isolate and investigate the effect of non-Gaussian wind velocity increments on wind turbine systems compared to Gaussian ones. The key aspects added to the ongoing scientific discussion are as follows:

- Firstly we emphasize the importance of isolating intermittency and discusses the requirements that wind fields must fulfil in the context of this problem.

- Secondly this work shows that an intermittency effect is evident for spatially homogeneous wind fields based on aero-servo-elastic wind turbine simulations. Preliminary results were reported in Ref. (Schwarz et al., 2018).

- Lastly, this work contributes to the discussion by pointing out the critical importance of the size of coherent structures and the general concept of coherence to this problem.

## 2 Methodology

To investigate the effect of intermittent vs. Gaussian wind, two synthetically generated wind field types are evaluated by means of wind turbine simulations and a subsequent load analysis. In the following, a comprehensive description into these synthetic wind fields is given in Sec. 2.1. The wind turbine model and the load analysis are discussed briefly in Sec. 2.2 and 2.3, respectively.

### 2.1 Wind fields

A central point of this work is the isolation of the intermittency effect. Hence, we searched for a method to generate wind fields with and without intermittency. Aside intermittency, these fields have the same features. They are equivalent according to the wind field characterization of the IEC standard. This is achieved by the utilisation of the CTRW model proposed by Kleinhans (Kleinhans, 2008), which is presented in detail in Sec .2.1.1.

The resulting temporal dynamics are discussed in detail in Sec. 2.1.2 alongside the requirements for the isolation of intermittency.

The isolation of intermittency could only be assured for individual post-processed CTRW time series $u(t)$, not for entire wind fields $u(y, z, t)$ as demanded in common wind turbine simulations. In order to compose entire wind fields from those time series, simplified spatial correlations in the $yz$-plain (parallel to th rotor disc) are considered. These are presented in Sec. 2.1.3.

General technical aspects about the wind fields are discussed in the following. Commonly ten minute wind samples are considered as the mean wind speed is assumed to be approx. stationary over this time span. Due to the high demand for data in order for the presented dynamics to be resolved reasonably well, stationary time series of the length of one hour are

considered in this study. For each type of field, ten realizations have been generated for each wind speed tested in this study. In order to investigate the impact on a model wind turbine profoundly, multiple mean wind speeds within the operation range of the turbine have been tested. These are $6\frac{m}{s}, 9\frac{m}{s}, 12.5\frac{m}{s}, 15\frac{m}{s}, 18\frac{m}{s}, 22.5\frac{m}{s}$ and $25\frac{m}{s}$. In order to obtain results independent from Turbulence Intensity (TI), TI is 10% in all cases. The sampling frequency of each data set is $f_s = 20$Hz. We assume the most relevant time scales of wind dynamics to the wind turbine system to be captured with this sampling frequency. As a consequence the smallest increment lag value within this study is $\tau_{\min} = \frac{1}{s_f} = 0.05$ sec. In the $yz$-plain a spatial discretionary of $31 \times 31$ equidistant grid points is used, spanning an area of $135m \times 135m$ covering the rotor area. This results in a mesh size of $dy = dz = 4.5m$, which is approx. the size of a discretized blade segment $dr$ in the utilized wind turbine model. For the sake of simplicity the fields are uniform meaning they do not consider shear, veer or similar aspects.

### 2.1.1 CTRW model

The model proposed by Kleinhans (Kleinhans et al., 2008; Kleinhans, 2008) is used for wind time series generation. It has been applied in previous studies related to the presented issue (Gontier et al., 2007; Mücke et al., 2011; Schwarz et al., 2018). Details are given in Appendix A. The models' main building blocks are two coupled Ornstein-Uhlenbeck processes and a stochastic mapping process, which are discussed in the following.

Velocity signals $u(s)$ are generated as coupled Ornstein-Uhlenbeck (OU) processes, cf. Ref. (Kleinhans et al., 2008), on a model-intrinsic time scale $s$. The resulting signal $u(s)$ is a stationary Gaussian process. The utilization and implementation of OU processes is widely known and thus not discussed further here (see Appendix A).

The key feature of the model is the stochastic time mapping process, which allows for the generation of intermittent dynamics. A mapping of the intrinsic time scale $s$ to the physical time scale $s \to t$ is realized as

$$\frac{dt(s)}{ds} = \tau_{\alpha,C}(s). \tag{2}$$

The mapping process $\tau_{\alpha,C}$ is essentially a waiting time distribution. This idea is based on the concept of Contious Time-Random Walks (CTRW), cf. e.g. Ref. (Kutner, Ryszard and Masoliver, Jaume, 2017). Kleinhans (Kleinhans, 2008) proposes a stochastic Lévy process for $\tau_\alpha$. For $0 < \alpha < 1$, $\tau_\alpha$ yields Lévy distributed random numbers larger than zero. In case of $\alpha = 1$ the mapping is linear $\tau_1 = 1$ so that $s = t$ and in return $u(s) = u(t)$. In order to avoid waiting times $\tau_\alpha \to \infty$ the Lévy process is bounded to yield a maximum waiting time $C$.

The intermittency of $u(t)$ is mainly determined by $\alpha$ and $C$. In this work, $C = 350$ sec and $\alpha_{\mathrm{int}} = 0.65$ are used for all intermittent cases. As mentioned above, $\alpha_{\mathrm{Gau}} = 1$ for Gaussian cases. A complete parametrisation is given in Appendix A.

### 2.1.2 Velocity time series for an isolated intermittency effect

The focus of this work is to assess the effect of different wind velocity increment statistics on wind turbine dynamics. For this purpose, wind time series with Gaussian and non-Gaussian (intermittent) increment statistics are generated and compared.

In order to investigate the impact of the increment statistics exclusively one must isolate them: The wind time series need to be highly comparable with respect to other, lower order statistics. For example: Differences between two time series with respect to the resulting wind turbine fatigue cannot be attributed strictly to the intermittency effect, if both time series differ not only in their increment statistics but also in other statistical quantities, say their standard deviations. To provide some reference, it was tested how sensitive the presented case is to changes in turbulence intensity. As a coarse overall trend it was found that a one percent gain led to a one percent increased fatigue load for the load sensors discussed in this study. This relationship is highly dependent on the actual turbine, the load sensor, other wind field etc. wherefore we do not claim this to be generally true.

Both resulting types of times series feature the same mean values and variances. Also the Probability Density Functions (PDFs) of the velocity fluctuations $u' = u - \overline{u}$ are equivalent. These qualify as One Point (1P) statistical properties. All moments of the 1P statistics will be equivalent for both types of fields. In addition the power spectrum which is related to the autocorrelation function via the *Wiener-Khinchin Theorem* is assured to be comparable. These statistics are Two Point (2P) statistics (alias increment statistics), more precisely 2P statistics of second order. The differences between both types of time series due to intermittency are evident in the fourth (and higher) moments of their 2P statistics. From turbulence theory it is known that the third moment of increments should also be differing between intermittent and Gaussian fields, as it is linked to the $\frac{4}{5}$ law of K41. This effect is not captured in our two types of time series as both of them feature non-skewed increment distributions. However, for the impact on loading however, we believe that the fourth moment is of much higher interest than the third moment. The third moment essentially determines the balance (or imbalance) between positive and negative wind speed increments. On the other side, the fourth moment similarly describes the balance (or imbalance) between large and small absolute amplitudes of the velocity increments. The amplitudes of wind speed increments are likely to correlate to some degree with the amplitudes of load cycles of a given load sensor. Therefore we focus on the fourth moment of the increment statistics.
In the following, features of the resulting time series are presented in detail.

First the 1P statistics are discussed: In order for the operation point of the wind turbine to be comparable the mean wind speeds have to be the same. The strength of wind fluctuations is scaled by the wind's variance, which is normalized by the mean wind speed is the TI, wherefore these have to be equivalent as well. Examples of time series for the main flow component $u(t)$ are given in Figs. 1a for the Gaussian type and 2a the intermittent type, respectively. It is evident that the mean wind speed and TI are equivalent between both types.

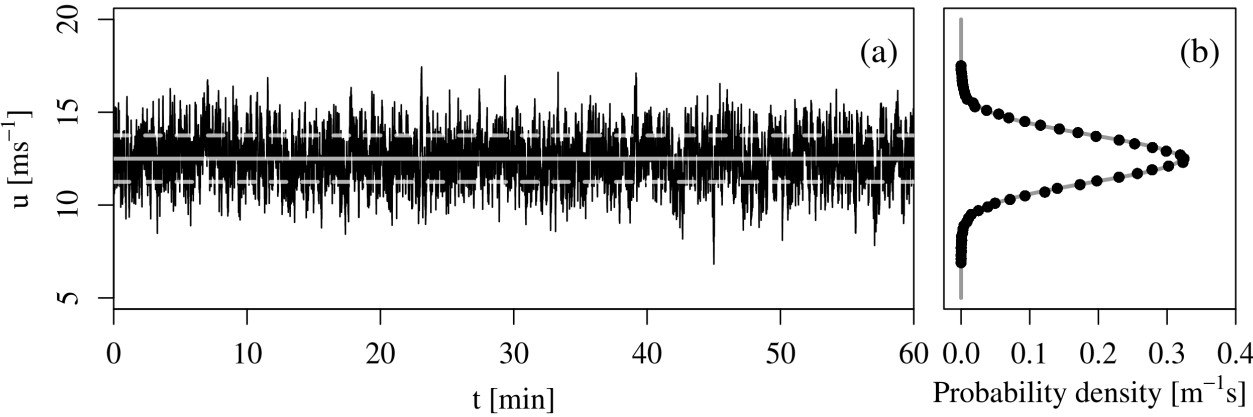

**Figure 1.** Examplary *Gaussian* wind time series. (a) Main wind velocity component $u(t)$, including mean $\pm$ 1 standard deviation (dashed lines). (b) Corresponding histogram and Gaussian fit.

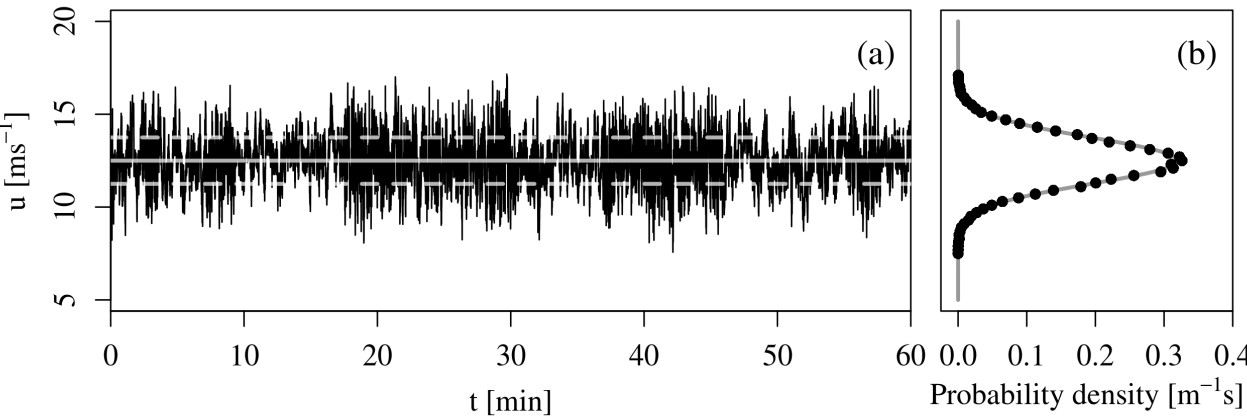

**Figure 2.** Examplary *intermittent* wind time series. Analogous to Fig. 1.

Furthermore, as shown in Figs. 1b and 2b, we achieve Gaussian velocity fluctuations $u'(t)$ for both types. It was assured that the first four central moments of the 1P statistics deviate from their desired values with by than $10^{-3}$. We are thus convinced that 1P PDF and 1P moments are comparable. We like to stress that within a study like this one, a high comparability of 1P statistics must be assured, before any observation in the results can be related to 2P statistics.

Next the 2P statistics of the generated time series are discussed. A common 2P statistic is the autocorrelation function

$$\rho(\tau) = \frac{E[u'(t)u'(t+\tau)]}{\text{VAR}[u']}. \tag{3}$$

It reflects how far wind structures extend in the longitudinal (in this case the temporal) dimension. Fig. 3 shows $\rho(\tau)$ for both, all intermittent and all Gaussian realizations. Aside from some scattering in the weakly correlated regime, a high agreement among
all realizations is evident: The velocity dynamics are correlated for roughly 12 seconds, which in comparison to atmospheric turbulence is very short. This can presumable be explained by the lack of low frequent dynamics in the velocity signal. It is evident from Figs. 1a and 2a that not too much low frequent dynamics present in our signals. The lack of those might potentially affect the presented results quantitatively, but not qualitatively, as the differences in the presented results stem from the intermittency. We were unable to incorporate more lower dynamics the velocity signals with our CTRW approach, as our
attempts spoiled other properties of our time signals. However it is important to keep in mind that it is our highest priority to work with highly comparable Gaussian vs. non-Gaussian fields. This compromised some of the other wind field parameters.

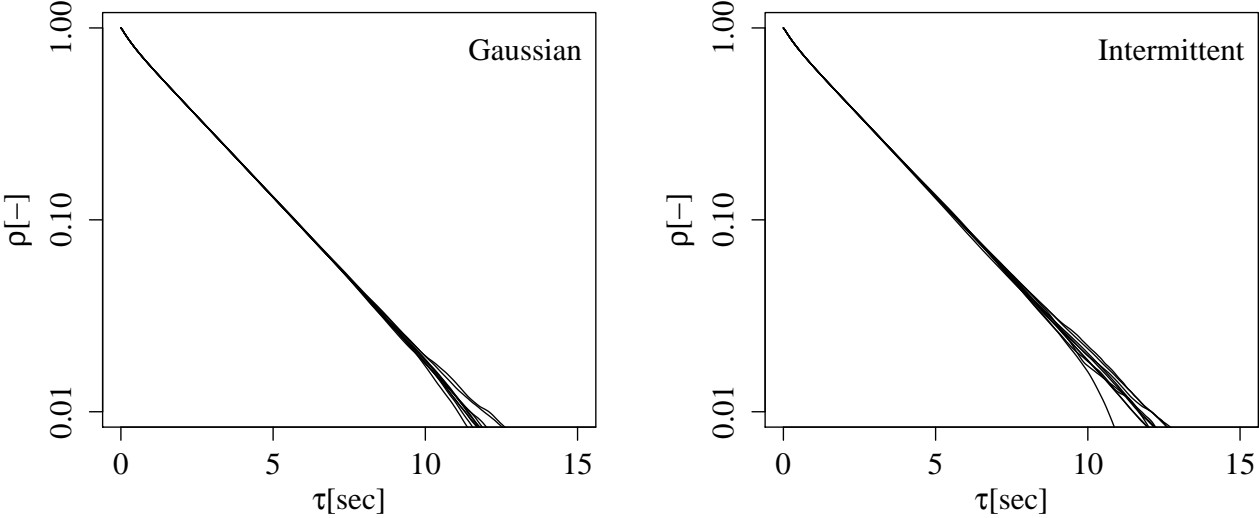

**Figure 3.** Correlation coefficient $\rho$ over lag value $\tau$ for ten Gaussian and ten intermittent realizations.

As mentioned above, $\rho(\tau)$ is related to the power spectrum via the *Wiener-Khinchin Theorem*. The power spectrum is commonly used to describe wind dynamics and has a known impact on the load dynamics, cf. Refs. (International Electrotechnical Commision; Veers, 1988). Fig. 4 shows the spectral properties of $u'(t)$ for both types of fields. Note that the intermittent
spectrum has been shifted vertically for the sake of better representation. As expected from the autocorrelations both types are described by highly comparable spectra. Also, within the frequency range $10^{-1} < f < 10^{0}$ the spectra roughly follow a

$-\frac{5}{3}$-trend, as postulated by Kolmogorov in 1941 (K41) (Kolmogorov, 1941).

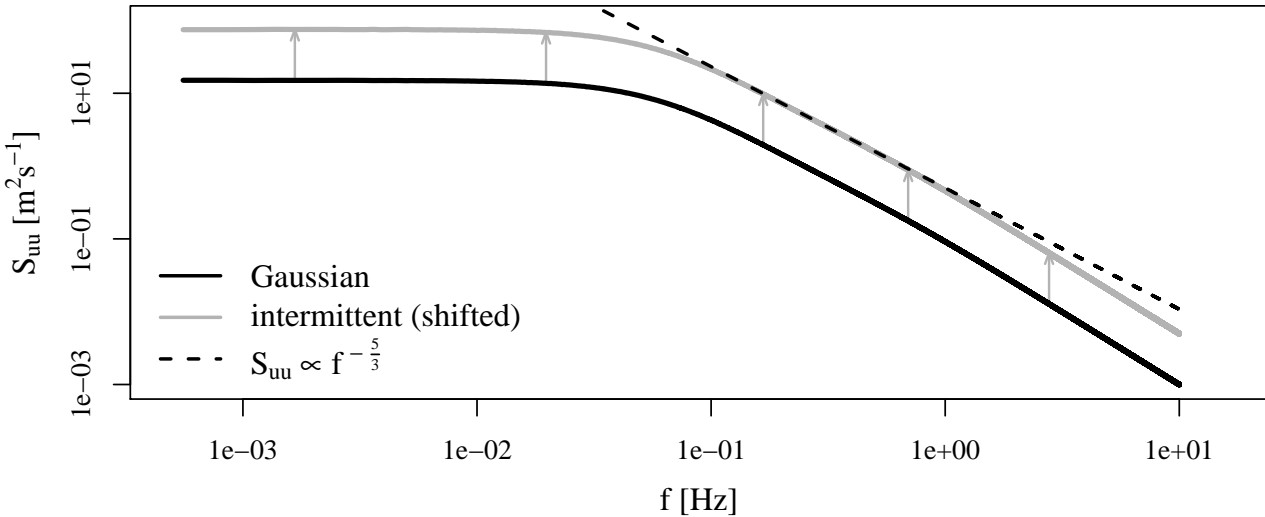

**Figure 4.** Power Spectral Density (PSD) $S_{uu}$ for the main wind velocity component $u(t)$ for the *Gaussian* and *intermittent* field type. Note that the intermittent spectra is shifted by a factor of five for better representation. The length of the arrows correspond to the distance of the shift. The dashed line represents a $-\frac{5}{3}$ slope relevant to K41.

The spectral and cross-spectral properties of the lateral and vertical velocity components are not discussed in this work. They have been modeled simplify as white noise signals, wherefore statistically significant differences between Gaussian and non-Gaussian fields are expected to arise from these velocity components. In general, these velocity components can drive fatigue loads for some sensors and turbine components. However in this work we focus on the impact of longitudinal wind dynamics and only discuss a limited set of load sensors.

Up to now, no differences between both types of wind fields are evident. This changes, when the PDFs of 2P statistics are considered: The non-Gaussianity of the 2P statistics is displayed in Fig. 5. Fig. 5a shows the histograms of wind velocity increments $\delta u_\tau$ for $\tau = \{0.05, 0.1, 2\}$. A deviation from an ideal Gaussian process is evident for the *intermittent* data set. The deviation is dependent on the time lag $\tau$ between the two points considered. In order to describe the scale dependency and the deviation from an ideal Gaussian PDF, Fig. 5b represents the fourth moment (alias flatness or kurtosis) of increment PDFs in dependence of $\tau$. As can be seen, the increment statistics of the *intermittent* data sets deviate from a Gaussian process in the range $\tau < 10s$.

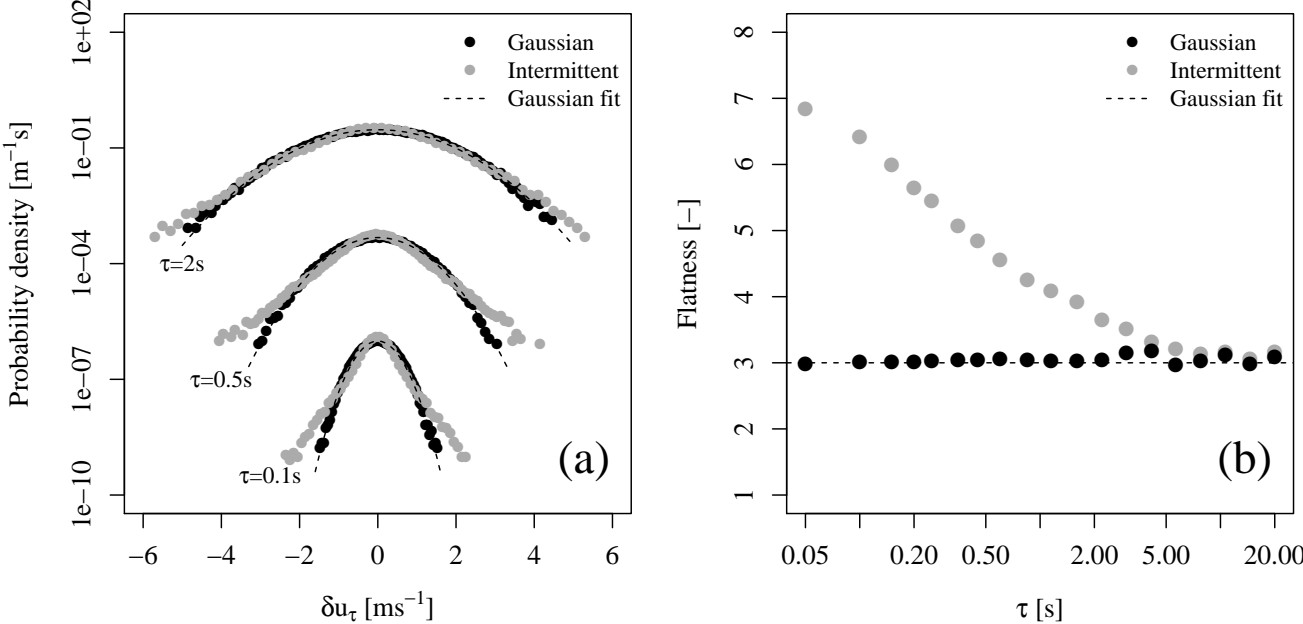

**Figure 5.** Comparison of the two-point statistics for the *Gaussian* and *intermittent* fields. (a) Histograms of velocity increment time series for $\tau = 0.1s, 0.5, 2s$, shifted vertically for better representation. (b) Flatness of increment PDFs in the range $0.05s \leq \tau \leq 15s$.

The fourth moment of the 2P statistics is the lowest order statistic in which differences between the two types of fields are evident.

### 2.1.3 Spatial dynamics

The statistical features presented in the previous section could be achieved for isolated velocity time series $\boldsymbol{u}(t)$, but not for
5   time series in a spatially correlated fields. In order to assemble velocity fields $\boldsymbol{u}(y, z, t)$, simplified spatial correlations are considered.

Firstly we generate a field, in which in all grid points the same time series is prescribed. This case is referred to, as the spatially *fully correlated* case. Secondly, a grid is filled with different, uncorrelated time series, which are independent from another, referred to as the spatially *delta correlated* case.

10   These are strongly idealized scenarios. Commonly, a spectrum of structure sizes is expected for realistic wind fields. For reference, integral length scales in the order of several hundred meters (Träumner et al., 2015) have been reported. We therefore argue that a realistic wind field at times may approach one or the other of these extreme scenarios and while not, is in between of them. In order to provide wind fields in between both of these two extreme scenarios, a subdivided $3 \times 3$ grid is considered. In the resulting nine sub-regions we prescribe fully correlated fields. These three cases are illustrated in Fig. 6.

The resulting correlations and coherence must be understood as a simplification of atmospheric turbulence, as it features a varying range of temporal and spatial scales. Since only regularized structures of one size are featured in the field, only the corresponding scales are contained in the fields. Implementing a more realistic coherence model into the CTRW model is very challenging: Coherence is typically incorporated by combining spectral properties of different grid points. However these

spectral properties correspond to the second moment of velocity increments via the *Wiener-Khinchin Theorem*. Therefore it is not possible or highly difficult to conserve all of the targeted properties of our wind time series when trying to implement coherence in the spirit of the Veers method (Veers, 1988). We believe this will have an impact on our results, mainly in a quantitative way: Coherence typically introduces the spatial variation in the wind field, which leads to the so-called 'eddy slicing' by the rotor, which in return is a driver for fatigue loads. This effect not fully captured in our approach, wherefore the

resulting load dynamics will feature less contributions for eddy slicing. However, we do not see the qualitative nature of our findings to be affected by these simplifications.

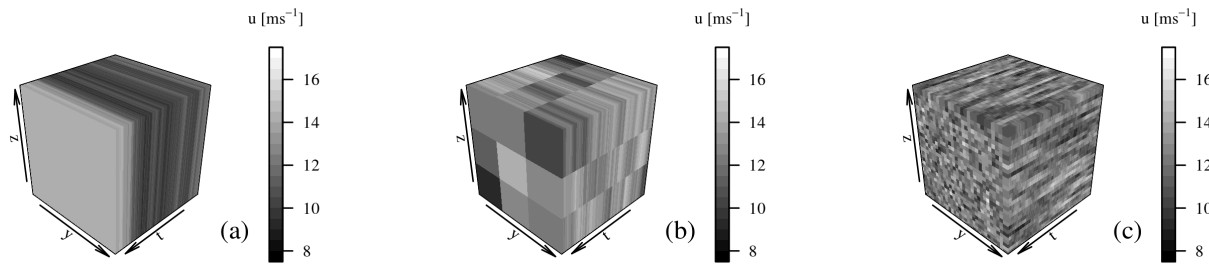

**Figure 6.** Exemplary visualization of the flow field excerpts for a mean velocity in main flow direction of $u = 12.5 \frac{m}{s}$. (a) *Gaussian* type, fully correlated in space. (b) *Gaussian* type, $3 \times 3$ subdivsion in space. (c) *Gaussian* type, delta correlated in space.

## 2.2   Wind turbine model

The wind fields generated in this study were applied in simulations of a common pitch regulated wind turbine. As a test wind turbine the well documented National Renewable Energy Laboratory (NREL) 5-MW Reference Wind Turbine, with a

rated wind speed of $11.4 \frac{m}{s}$ was selected (Jonkman et al., 2009). Aero-servo-elastic simulations were carried out using *FAST* (Jonkman and Buhl, 2005; Jonkman and Jonkman, 2016) (v8.15), including its BEM code *AeroDyn15* (v15.02) (Jonkman et al., 2016). Active pitch and variable speed control, as well as common add-ons to the pure BEM model for e.g. tip-losses or the tower effect etc. are taken into consideration by application of typical correction models.

BEM theory is based on stationary airfoil data and further on an equilibrium wake (Schepers, 2012). Dynamic airfoil be-

haviour is modelled with a Beddoes and Leishman type of dynamic stall model with minor code specific modifications (Leishman and Beddoes, 1989; Jonkman et al., 2016). Generally, it is an open question how well load dynamics resulting from turbulent inflow are captured by a BEM method. However, Madsen et al. (Madsen et al., 2018) recently presented evidence that BEM codes are capturing general trends well.

## 2.3 Load analysis

The resulting wind fields are analysed with respect to the load response of a wind turbine model. The so obtained load time series can be analysed in a manifold of ways. In this work we focus on fatigue loads as these are dependent on the entirety of the wind and load statistics. A specific methodology is proposed by certification guidelines, such as Ref. (International Electrotechnical Commision) for this kind of load analysis. Following Ref. (International Electrotechnical Commision) a Rain-Flow-Counting (RFC) procedure (Matsuishi and Endo, 1968) is conducted. In doing so a load history is simplified into a sequence of local load maxima. Subsequently so called load ranges $r_i$ are calculated from this sequence. Load ranges are essentially load increments similar to Eq. 1, but calculated exclusively between local load extrema. The resulting load ranges $r_i$ are utilized to calculate an Equivalent Fatigue Load (EFL)

$$\text{EFL} = \left( \frac{1}{T} \sum_{i=1}^{k} r_i^m \right)^{\frac{1}{m}}, \tag{4}$$

where $r_i$ is the $i$-th load range and $T$ represents the number of seconds covered by the load history. Due to the selection of $T$ this specific load is referred to as the 1Hz EFL. Further, $m$ is the SN-slope coefficient (in this study $m_{\text{Tow}} = 4$ for the tower, $m_{\text{DrvTr}} = 8$ for drive train components and $m_{\text{Bld}} = 12$ for the blades). Intuitively, the EFL can be understood as the peak-to-peak amplitude of a hypothetical load cycle with the period of $1s$, which leads to the same damage accumulation as the input load history over the time $T$.

Note that in practice, the EFL is calculated over a wide range of wind speeds. The wind specific fatigue loads are subsequently combined with a wind probability distribution and integrated up so that finally one value for the entire lifetime is obtained. In this work, we focus on the loading at specific wind speeds in order to identify potential trends.

## 3 Results & Discussion

Due to the limited scope of this paper not all turbine components and load sensors can be discussed here. We focus mainly on load sensors that are expected to be responsive to wind dynamics and not dominated by other effects, such as gravitational forces. The two sensors presented here exemplarily are the rotor thrust and the Tower base Bending Moment Fore-Aft (TBMFA).

This section is organized as follows: Firstly, the fully correlated wind field case is discussed. Subsequently, this result is compared against the results from the delta correlated case and the intermediate $3 \times 3$ case is discussed at last. Finally, 2P load statistics for all cases are compared to underline the impact of the spatial dynamics and the coherence of the wind field.

## 3.1 Results for fully correlated fields

Figs. 7 and 8 show the comparison of EFLs obtained for the *Gaussian* and *intermittent* fields for the *fully correlated* case in absolute and relative values for the rotor thrust and Tower Base bending Moment Fore-Aft (TBMFA). Differences between *intermittent* and *Gaussian* types for both sensors are evident in case of *fully correlated* fields at all wind speeds. As a rough overall trend, the intermittent cases seem to yield a 5 to 10% increase in fatigue loads, which can be relevant in a design or cerification process. Due to the isolation of intermittency discussed in Section 2.1.2, this difference can directly be associated to the intermittent statistics.

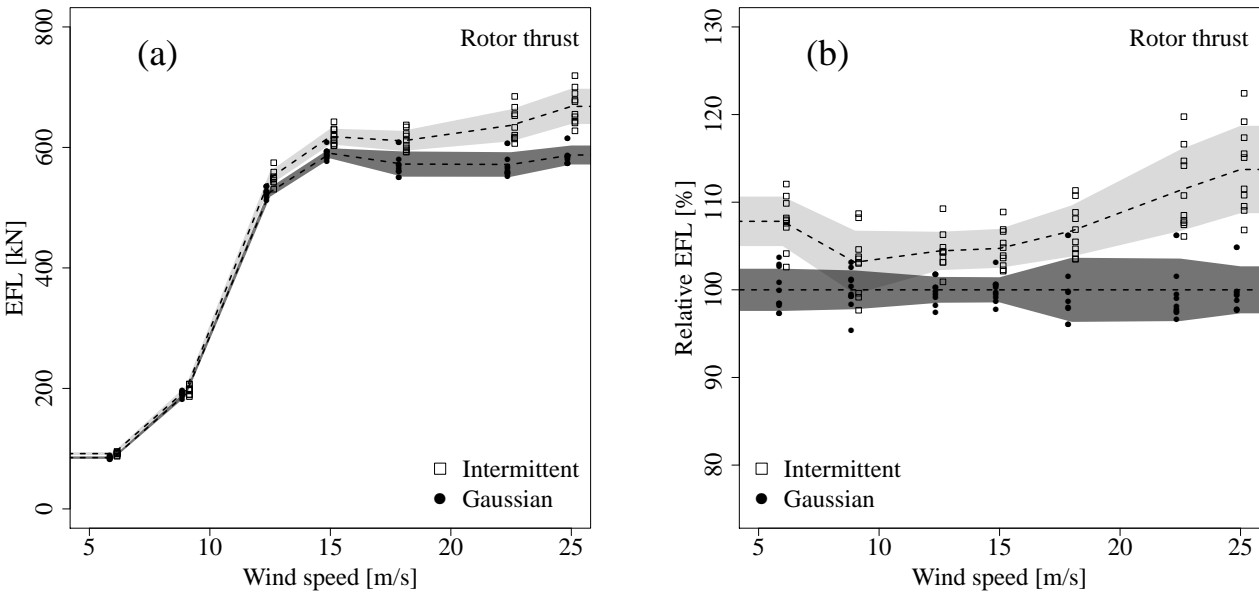

**Figure 7.** EFL (cf. Eq. (4)) for the rotor thrust for all realizations and all wind speeds for the *fully correlated* case. The data points were shifted horizontally for a better distinction between *intermittent* and *Gaussian* data. The dashed lines represent averages, the shaded area covers ± one standard deviation around the average. (a) Standard represenation. (b) Data normalized with the average of the *Gaussian* result (Gaussian averages correspond to 100%).

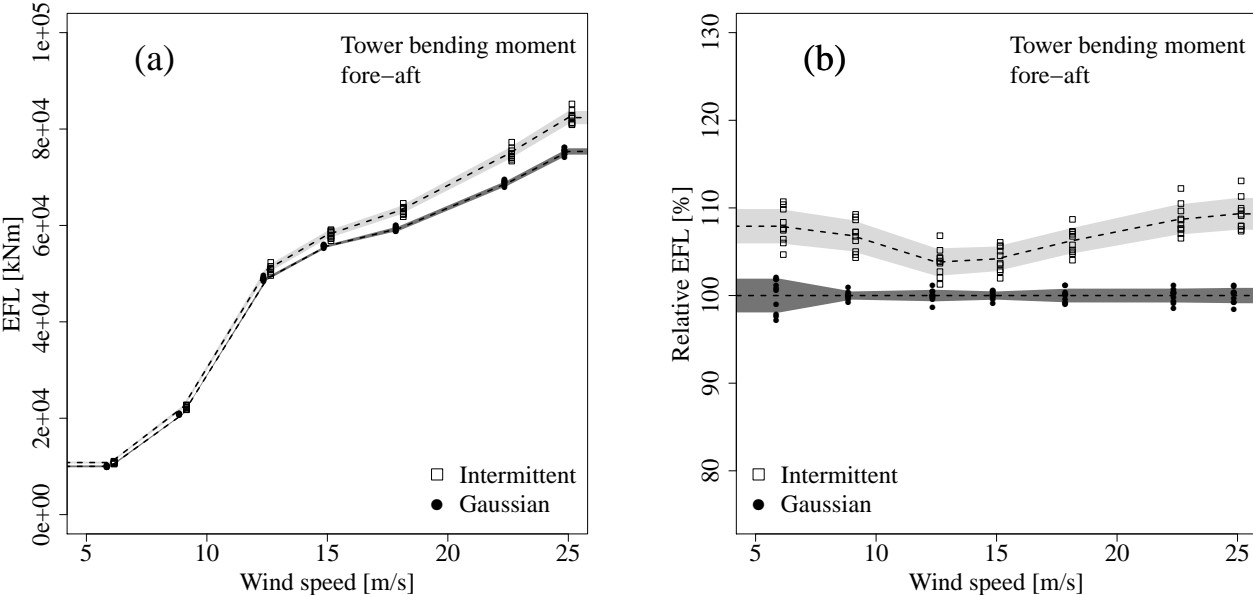

**Figure 8.** EFL for the Tower base Bending Moment Fore-Aft (TBM FA). Analogous to Fig. 7.

## 3.2 Results for delta correlated fields

Next, the delta correlated wind fields are considered. Figs. 9 and 10 show the comparison for the *delta correlated* case for the rotor thrust and Tower Base bending Moment Fore-Aft (TBMFA). It is evident that there is no significant difference between the Gaussian and intermittent wind fields. Physically, this behaviour can be explained as the blades are excited by a multitude of uncorrelated processes over the course of a rotor revolution. The dynamics of rotational sampling of the spatial variations outweigh the effect of the temporal statistics.

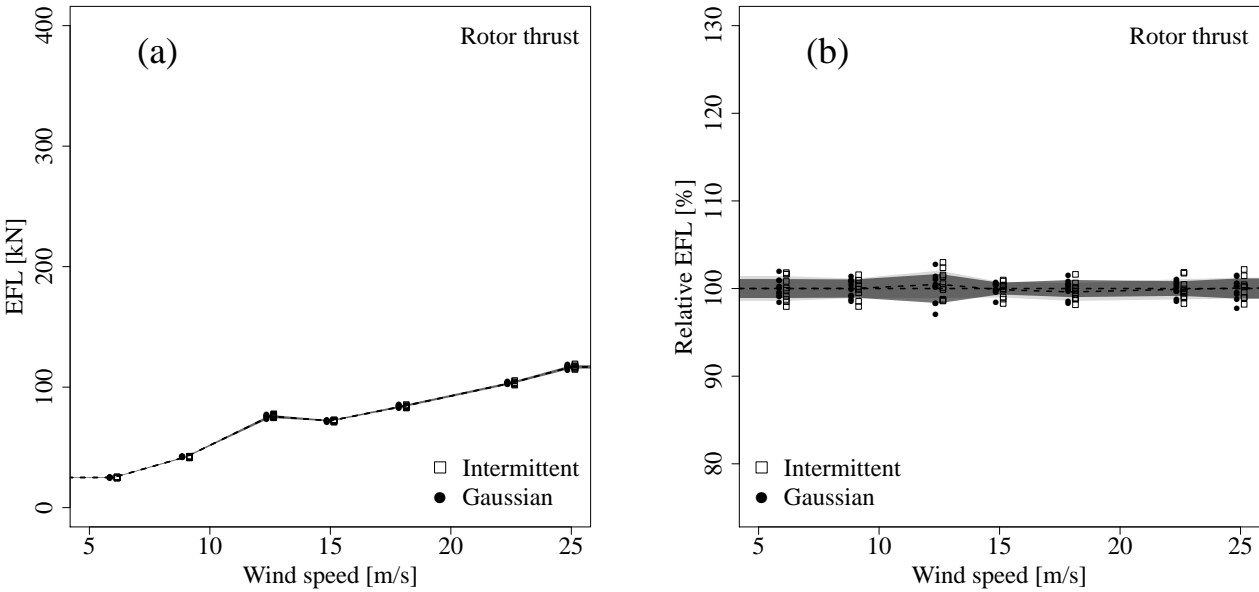

**Figure 9.** EFL for the rotor thrust for the *delta correlated* case. Analogous to Fig. 7.

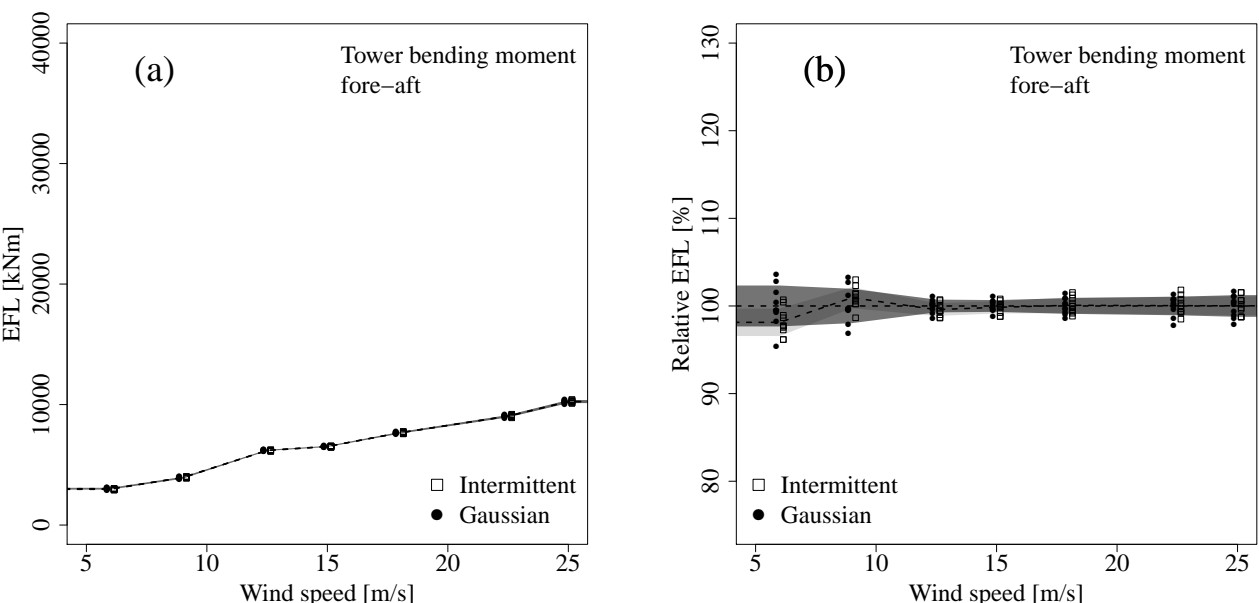

**Figure 10.** EFL for the Tower base Bending Moment Fore-Aft (TBM FA) for the *delta correlated* case. Analogous to Fig. 7.

### 3.3 Results for subdivided 3 × 3 field

Finally, an intermediate case between the fully and delta correlated wind fields is presented. Results are shown in Figs. 11 and 12. For both sensors still differences between the Gaussian and intermittent cases are evident. However these are much less pronounced than for the fully correlated case. In contrast to the delta correlated case however, differences can be identified.

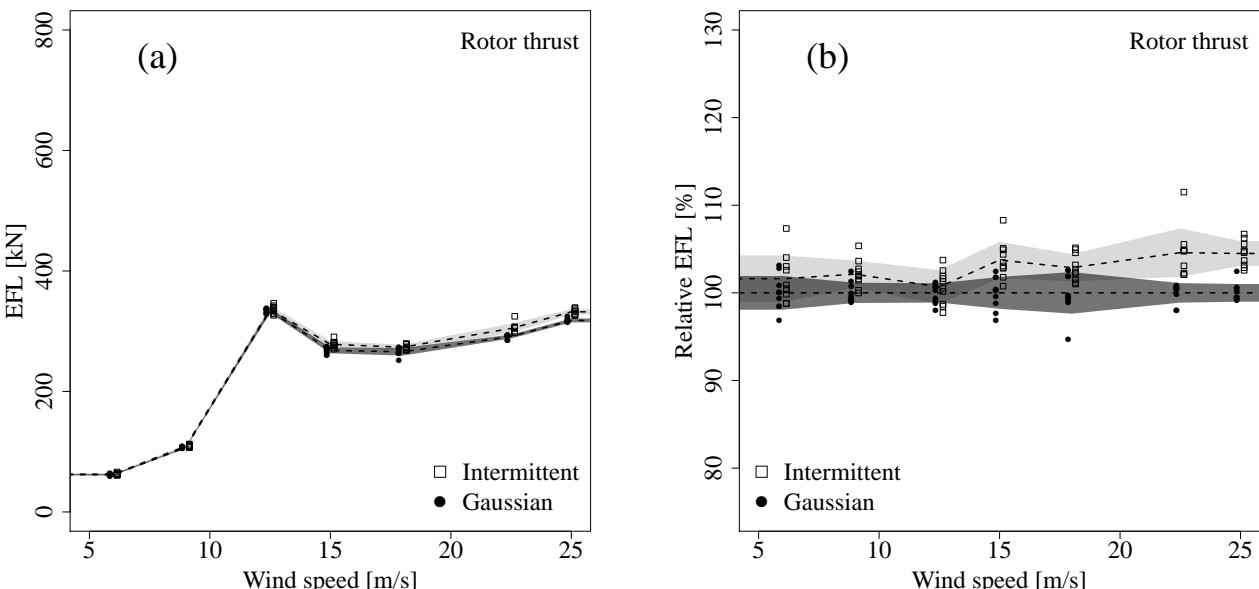

**Figure 11.** EFL for the rotor thrust for a subdivided 3 × 3 field. Analogous to Fig. 7.

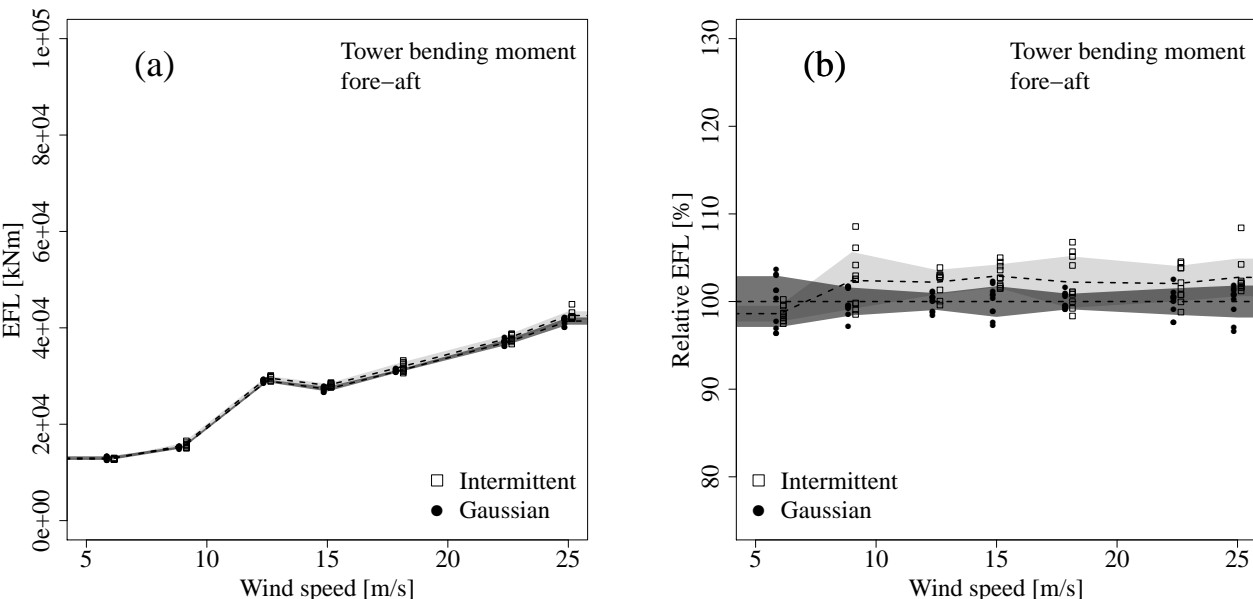

**Figure 12.** EFL for the Tower base Bending Moment Fore-Aft (TBM FA) for a subdivided $3 \times 3$ field. Analogous to Fig. 7.

### 3.4 Load increment statistics

In order to show to which degree the intermittent wind dynamics affect the load dynamics in each of the three types of wind fields, the statistics of the load increments $x_\tau$ (cf. Eq. (1)) are discussed. In this context $x$ denotes any load sensor. Fig. 13 shows the flatness of the resulting rotor thrust and TBMFA increments at $12.5 \frac{m}{\sec}$.

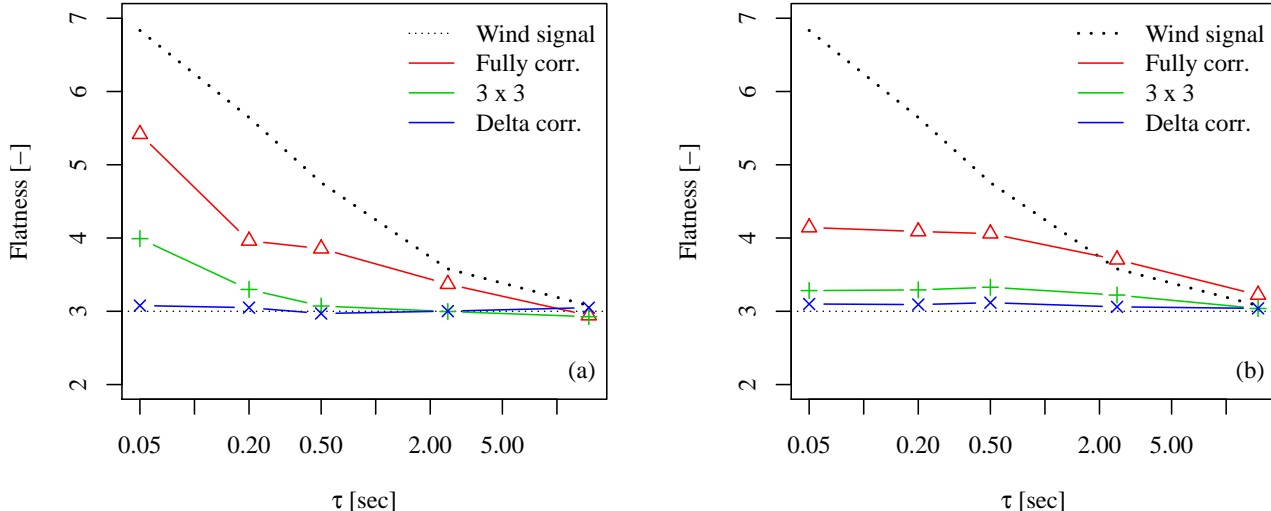

**Figure 13.** Flatness of load increment PDFs for intermittent fields with different spatial dynamics. (a) For the rotor thrust. (b) For the Tower base Bending Moment Fore-Aft (TBM FA).

It is evident that for the fully correlated case, the non-Gaussianity of the wind signal is induced into the rotor thrust. Deviations from the wind statistics of $u_\tau$ can be explained by the complex and non-linear aero-servo-elastic response of the rotor thrust signal to the incoming wind dynamics. These are individual for each load sensor as can be seen when comparing the thrust signal with the TBMFA signal in Figs. 13a and 13b, respectively.

For the delta correlated case the load signal become quasi perfectly Gaussian (flatness $= 3$). The Gaussianity of these load dynamics can be explained with the *Central Limit Theorem*: The rotor is excited by $31 \times 31 = 961$ uncorrelated random variables over the course of one revolution. The superposition of all of these processes yield a regular and steady Gaussian dynamic.

Lastly the $3 \times 3$ subdivision is considered. Comparable to the EFL results, the load increments for this field type features some remaining non-Gaussianity, however much less compared to the fully correlated case.

At this point we need to recall that all three spatial compositions are idealizations. A real wind field features structures up to length scales of several hundred meters (Träumner et al., 2015), as the fully correlated case. However it also features structures of small size as in the delta correlated case. However it will contain more than one structure size. It remains thus an open question, which cases represents real wind fields most adequately.

## 4 Conclusions

This work adds to the discussion of intermittency in the context of wind turbine dynamics as follows:

- In order to evaluate the impact of higher order statistics or intermittency in wind dynamics they need to be isolated properly. An example for such an isolation has been given in this work.

- Intermittent wind dynamics and advanced wind statistics can be relevant to the fatigue loads of wind turbines. This holds true for an aero-servo-elastic wind turbine response including pitch control and variable rotor speeds. An effect on the purely aerodynamic response has been reported in Ref. (Schwarz et al., 2018).

- The quantitative values presented in this work have to be treated with caution due to the simplified spatial correlations in the wind fields. Still, to provide a quantitative estimate the lifetime fatigue has been calculated based on a wind class III Weibull distribution (shape = 2, scale= 8.46). The relative increase in fatigue due to intermittency for the rotor thrust equates to 105.7% (fully correlated), 102.3% ($3 \times 3$ case) and 100.2% (delta correlated).

- The intermittency effect depends on the size and number of wind structures, also referred to as the coherence of the field. Highly coherent fields show an intermittency effect, incoherent fields do not. The dependence on spatial variation needs to be investigated further.

## Appendix A: CTRW model parametrisation

For details and other deployments of this model the reader may be referred to Refs. (Kleinhans et al., 2008; Kleinhans, 2008; Mücke et al., 2011; Gontier et al., 2007). Time series generation in the CTRW model bases on two coupled Ornstein-Uhlenbeck (OU) processes and a stochastic mapping.

The OU processes are

$$\frac{d\boldsymbol{u}_r(s)}{ds} = -\gamma_r \left(\boldsymbol{u}_r(s) - \boldsymbol{u}_0\right) + \sqrt{D_r}\boldsymbol{\Gamma}_r(s) \tag{A1}$$

and

$$\frac{d\boldsymbol{u}_i(s)}{ds} = -\gamma \left(\boldsymbol{u}_i(s) - \boldsymbol{u}_r(s)\right) + \sqrt{\boldsymbol{D}_i}\boldsymbol{\Gamma}(s). \tag{A2}$$

In the previous Equations

$$\boldsymbol{u}_r(s) = \begin{pmatrix} u_r^{(x)}(s) \\ u_r^{(y)}(s) \\ u_r^{(z)}(s) \end{pmatrix}, \quad \boldsymbol{u}_0(s) = \begin{pmatrix} u_0^{(x)}(s) \\ u_0^{(y)}(s) \\ u_0^{(z)}(s) \end{pmatrix} \quad \text{and } \boldsymbol{\Gamma}_r(s) = \begin{pmatrix} \Gamma_r^{(x)}(s) \\ \Gamma_r^{(y)}(s) \\ \Gamma_r^{(z)}(s) \end{pmatrix}. \tag{A3}$$

$\Gamma$ represents white noise.

The stochastic mapping process $s \to t$ is realized as

$$\frac{dt(s)}{ds} = \tau_{\alpha,C}(s), \tag{A4}$$

where $\tau_{\alpha,C}$ is a bounded stochastic process yielding Lévy distributed random numbers in the range $0 < \tau_{\alpha,C} < C$. Note $\tau_{1,C} = 1$ so that $s = t$ and $\boldsymbol{u}_i(s) = \boldsymbol{u}_i(t)$. An implementation of such a process can be achieved as

$$\tau_\alpha = \frac{\sin\left(\alpha\left(V + \frac{\pi}{s}\right)\right)}{\cos(V)^{\frac{1}{\alpha}}} \left(\frac{\cos\left(V - \alpha(V + \frac{\pi}{2})\right)}{W}\right)^{\frac{1-\alpha}{\alpha}}, \tag{A5}$$

with V being uniformly distributed random variable between $\left[-\frac{\pi}{2}, \frac{\pi}{2}\right]$ and W an exponential distribution with unit mean.

Constants:

$$u_0^{(x)} = \{6, 9, 12.5, 15, 18, 22.5, 25\} \frac{m}{\sec} \tag{A6}$$

$$u_0^{(y)} = u_0^{(z)} = 0 \frac{m}{\sec} \tag{A7}$$

$$\gamma \approx 1.6595\omega_S \tag{A8}$$

$$\gamma_r \approx 0.2150\omega_S \tag{A9}$$

$$D_r = 0.1921\omega_S\sigma^2 \tag{A10}$$

$$D_i = 0.3468\omega_S\sigma^2 \tag{A11}$$

$$\omega_S = 1.8\frac{1}{\sec} \tag{A12}$$

$$\sigma = 0.1u_0^{(x)} \tag{A13}$$

$$C = 350\sec \tag{A14}$$

$$\alpha_{\text{Gau.}} = 1 \tag{A15}$$

$$\alpha_{\text{int.}} = 0.65 \tag{A16}$$

*Author contributions.* C.M. Schwarz generated the wind fields, conducted the simulations and the load analysis and wrote the manuscript.

15 S. Ehrich refined the wind model and was equally involved with the generation of the wind fields and interpretation of the results. J Peinke had the initial idea, supervised the work and helped preparing the manuscript.

*Competing interests.* No competing interests are present.

*Acknowledgements.* Parts of this study have been carried out within the AVATAR project, which has received funding from the European Union's Seventh Programme for research, technological development and demonstration under grand agreement No FP7-ENERGY-2013-1/no 608396. The authors would also like to thank the editor and referees. Their comments were very helpful in improving this manuscript.

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
