# Peer review of "Wind turbine load dynamics in the context of turbulence intermittency"

_Wind Energy Science, 2019_

## Referee Comment (RC2)

[referee-annotated manuscript omitted]

---

## Referee Comment (RC1) · Jacob Berg (Referee) · 26 Jun 2019

Comment on "Wind turbine load dynamics in the context of turbulence intermittency"

Jacob Berg, DTU Wind Energy (Referee) jbej@dtu.dk

Addressing the non-gaussian behavior of small-scale turbulence upon wind turbine fatigue loads (through an aero-servo-elastic wind turbine model) are investigated with a simple Continuous-time-random walk model. The focus is on spatial correlations which are added through idealized prescribed correlations. The focus of the paper is clear (except the part explaining the spatial correlation – see below) and the methodology by Berg et al. 2016 in which the small-scale non-gaussian behavior (as defined though velocity increments) was isolated is to a large degree followed. The paper thus successfully bridges the more classical approach by using continuous-time-random walk models without spatial correlations (see papers in reference list mainly from the Oldenburg group by J. Peinke) and the work by Berg et al. The literature in the field is nicely reviewed and strength and weaknesses in both approaches are highlighted. The conclusions that the effect of intermittency decreases as spatial correlations increases is indeed interesting and worth investigating even further. This is therefore also my main critic of the paper – exactly how is this correlation constructed? and how is the associated coherence related to atmospheric turbulence.

I have added specific comments below for the authors to correct / answer / reflect on.

1.  P1,l3-6: which two types? You just say two types of wind fields. Please specify

2.  P1,l3. In IEC – the Kaimal model must be accompanied by a coherence model to account for the spatial correlation – much in connection to your work.

3.  P6,l25. Can you say something about the physical time scales in the wind model (here defined with constants of physical units). How does for example the integral timescale compare to characteristic time scales of velocity fluctuations in the atmospheric boundary layer like ~10min.

4.  P7,l5. 1% error in standard deviations of the wind velocity leads to approx. 1% in fatigus loads . Can you please elaborate – or add reference. So a linear response? – and for all load channels?

5.  P7,l13. In principle also in the third moment (zero in gaussian stat).

6.  P8,l1. In general, for most figures (fig 1b and 2b, 3 and 4) – if you want to compare two curves, in this case two pdf of velocity, u'(t), then plot the curves in the same coordinate system. Makes it much easier to observe any likely differences.

7.  Figure 5a. Is it possible to get the numbers on the x-axis as a function of standard deviations ($\sigma_{\tau}$). That makes it easier to judge how much non-gaussian the velocity increments at the different time delays are.

8.  Section 2.1.3: Main question: How are the delta correlated and 3*3 block correlated cases constructed? A 2-3-line description is not enough. For example, what is the block size in physical units and hence the derived integral length scale? Also how does the Coherence (based on the cross spectrum) look and how does it compare to atmospheric numbers and for example the Davidson model (exponential decay)? Both text, formulas and Figures are required. This is a vital information since the novel-ness of this paper is exactly the including of spatial correlation to the already introduced and utilized Continuous-time-random walk model.

9.  Figure 14 caption: "vertically" should be "Horizontally"

10. Figure 7-12. How many ensemble members is used (difficult to count the number of boxes in the plots) and what is the size in space and time of these turbulence boxes.

11. A discussion of your results in terms of turbine safety factors would make the significance of the results more useful.

---

## Author Comment (AC1) · 28 Jul 2019

Dear Jacob Berg,

thank you very for your remarks and comments on the manuscript as well as for the discussions at the conferences in the past years.

One could say that some remarks targeting e.g. representation are a question of taste. In these cases we respectfully preferred to stick to the original manuscript.

*Comment from Referee 1:*
*Exactly how is this correlation constructed?*

*How is the associated coherence related to atmospheric turbulence?*

Authors response:

*The focus in this work is on the temporal statistics of the wind. We were able to generate time series with very specific properties and faced the challenge to assemble wind fields based on these very time series.*

In one of the extreme cases we prescribe one and the same time series in every of the wind fields grid points in the rotor plane (fully correlated). In another extreme case (delta correlated) we prescribe completely independent time series in every of these grids points, which results in an uncorrelated or delta-correlated field (delta correlated). In order to provide wind fields in between both of these two extreme scenarios, we designed fields in which sub-regions of the grid in the rotor plane are defined, in which we prescribe fully correlated fields. For example, the grid can be sub-divided into 9 regions of a thee by three grid. In these regions the exact same wind time series will be prescribed.

The resulting correlations and coherence must be understood as a simplification of atmospheric turbulence, since atmospheric turbulence features a varying range of temporal and spatial scales.

*Comment from Referee 1:*
*P1,l3-6: which two types? You just say two types of wind fields. Please specify*

Authors response:

Gaussian and non-Gaussian

Modifications to the manuscript:

This has been edited in the manuscript.

*Comment from Referee 1:*
*P1,l3. In IEC – the Kaimal model must be accompanied by a coherence model to account for the spatial correlation – much in connection to your work.*

Authors response:

There seems to be typo with respect to line you indicate.

We mention the Kaimal model in the introduction as one of the examples for a wind model listed in design guidelines. It is true and we are aware of the fact that the Kaimal model targets the spectral properties and is used in combination with a coherence model. However we are of the opinion it is sufficient at this part of the manuscript to simply refer to wind fields constructed in such a fashion as „Kaimal wind fields" or based on the „Kaimal model".

*Comment from Referee 1:*
*P6,l25. Can you say something about the physical time scales in the wind model (here defined with constants of physical units). How does for example the integral timescale compare to characteristic time scales of velocity fluctuations in the atmospheric boundary layer like ~10min.*

Authors response:

We believe some of the aspects you are asking for are addressed in section 2.1.2. The time series are correlated for roughly 12 seconds, which in comparison to atmospheric turbulence is very short. An integral time scale was not calculated. In our opinion the reason for this is a lack of low frequent dynamics in the velocity signal. As can be seen in e.g. Fig. 1 & 2: There are not too much low frequent dynamics present in our signals. The lack of those might potentially affect the presented results quantitively, but not qualitatively, as the differences in the presented results stem from the intermittency.

We tried to incorporate lower dynamics the velocity signals with our CTRW approach, but essentially found that it would have spoiled other properties of our time signals. Important is to keep in mind that it is our highest priority to have highly comparable Gaussian vs. non-Gaussian fields. We therefore had to trade-off some wind field properties.

*Comment from Referee 1:*
*P7,l5. 1% error in standard deviations of the wind velocity leads to approx. 1% in fatigus loads .*
*Can you please elaborate – or add reference. So a linear response? – and for all load channels?*

Authors response:

We were aiming to provide some reference for the reader how a 1% change in fatigue load can be understood.

The idea was to give a quantitative estimate on how big an offset in turbulence intensity must be, in order to provoke a 1% difference in fatigue loads. We conducted some test simulations with our set-up and found that a 1% increase in turbulence intensity yields roughly a 1% increase in fatigue loading. This is of course highly dependent on the actual turbine, the load sensor, other wind field etc. This is why we added a footnote to emphasize that this result is simply a rough estimation.

*Comment from Referee 1:*
*P7,l13. In principle also in the third moment (zero in gaussian stat).*

Authors response:

Both of our wind signals (the Gaussian and the non-Gaussian) feature zero skewness. Strictly speaking this is another simplification and deviation from atmospheric turbulence, however it is the aim of this work to focus on the fourth moment and to have no differences in any of the lower order statistics, including the third moment of the 2P statistics.

*Comment from Referee 1:*
*P8,l1. In general, for most figures (fig 1b and 2b, 3 and 4) – if you want to compare two curves, in this case two pdf of velocity, u'(t), then plot the curves in the same coordinate system. Makes it much easier to observe any likely differences.*

Authors response:

Thanks for this advice. Aside for the spectrum we decided to have two plots and make them comparable by having the exact same scaling. Since the content is so much alike both graphs would practically be on top of each other, which in return would also hamper visibility. If you would put e.g. both time series in one plot, this would make it very busy.

*Comment from Referee 1:*
*Figure 5a. Is it possible to get the numbers on the x-axis as a function of standard deviations*
*(\sigma_{\tau}). That makes it easier to judge how much non-gaussian the velocity increments at*
*the different time delays are.*

Authors response:

It is indeed common to normalize the x-axis by the respective standard deviation.

However, we are of the opinion it is much more intuitive for readers who are new to this topic to understand the actual increments in m/s. For instance, it becomes evident from Fig. 5a that there are no velocity increments with an amplitude of the value 2 m/s in the Gaussian signal, whereas there are some of these events in the non-Gaussian signal. We are of the opinion this provides a more intuitive understanding of the actual velocity differences.

For the purpose of judging / quantifying the non-Gaussianity Fig. 5b is provided, which shows the fourth moment of the velocity increments.

---

## Author Comment (AC2) · 28 Jul 2019

Dear Vasilis Riziotis,

thank you very much for your comments and thoughtful questions.

Thank you as well for the careful remarks on spelling and syntax, which are highly appreciated. All of your remarks related to the wording of the manuscript have been edited. They will not be listed here.

*Comment from Referee 2:*
*The main weak point of the analyses is that between the two unrealistic extreme scenarios a) of the fully correlated and b) the uncorrelated wind fields there is no strong evidence that the intermediate field is close (in terms of spatial coherency) to the conventional wind field, against which load predictions are compared. A way to mitigate the above ambiguity is to compare spatial coherence function of the two fields like the authors do with autocorrelation function. At least this will give the picture of how realistic is the about 5% increase on the fatigue loads pre- dicted when intermittency effect is taken into account.*

Authors response:

We agree that the coherence of any of the presented wind fields is highly idealized and that this indeed hampers the transferability of the presented results to practical applications. We hope this is clear by showing visual representations of the fielda in Fig. 6. However the main message of this paper is not that the intermittency will lead to an increase in fatigue by 5%, but rather that it can be demonstrated that the results are indeed affected, when intermittency is considered. The considered statistics do not seem to be filtered out by the rotor. In conclusion this work is more fundamental than applied research.

Unfortunately we did not manage to incorporate a more realistic coherence model within the time frame of this project due to the complexity of the problem. Our aim was to generate wind fields, featuring highly comparable statistics aside for the intermittency, more precisely the fourth moment of the 2 point statistics (and of course even higher statistics). We were abled to achieve this for individual time series only, not for entire fields. Thus, we faced the challenge to assemble wind fields from these time series without changing them and this simple approach was what we came up with first.

In related work we experimented with wind fields featuring more realistic spatial dynamics, but failed to conduct a consistent comparison between Gaussian and non-Gaussian. Briefly said the spatial dynamics had a bigger impact on the results than the temporal dynamics, which hampered drawing conclusions.

Note that intermittency is also expected to occur in the spatial dimensions. In the end it comes down to the lack of a model for turbulence that incorporates everything at once.

Modifications to the manuscript:
The conclusion section was edited to clearify this point.

*Comment from Referee 2:*
*Figure 4 presents spectral characteristics of the axial wind component. Do you get the same good agreement for the other components. Moreover, the ratios sdv_u/sdv_v and sdu/sdv_w are they also maintained? These are also important parameter that drive fatigue loads.*

Authors response:

The spectral and cross-spectral properties of the lateral and vertical velocity components are not discussed in this work. They have been modeled simplify as white noise signals, wherefore we do not expect any statistically significant differences between Gaussian and non-Gaussian fields to arise from these velocity components. Again be reminded that we value comparability between non-Gaussian and Gaussian fields over „realistic-ness".

*Comment from Referee 2:*
*The authors compare equivalent loads of the thrust and tower bottom bending moment. The above two load sensors are pretty much correlated and therefore they do not offer any additional information the one with respect to the other. It would be preferable to compare blade flapwise moment (which corresponds to the rotating frame) and tower top yaw and tilt moment plus the thrust or tower bottom bending moment. Yawing and titling moments are much more sensitive to the incoherent nature of the inflow.*

Authors response:

Some of the sensors you indicate have indeed been analyzed in the scope of this work. The two sensors presented in this manuscript were the ones in which the trends were most clear. While they are as you point out highly correlated, still slight differences between their results are evident.

Other sensors, such as the blade root moment out of plane or the rotor torque did feature similar differences between Gaussian and non-Gaussian fields in the same magnitude. However their individual responses were not straightforward to explain. We took an in-depth look into their dynamics, but decided that for the message of this paper, a small set of sensors would be sufficient, as we wanted to avoid lengthy discussions of load dynamics that are not related to intermittency.

In other words, we show here the sensors that could be analyzed and understood the easiest.

*Comment from Referee 2:*
*A global, lifetime fatigue damage estimation could be provided based on a standard Weibull or Rayleigh distribution of the wind.*

*It is recommended to extend the conclusion section by adding some qualitative discussion of the predicted change in the fatigue loads.*

*It would be nice to provide some quantification in the conclusions section.*

Authors response:

We are aware that in a real application problem the wind-specific equivalent loads are combined with the probability of its wind bin and then integrated up. We calculated the relative EFL based on a wind class III Weibull distribution for the thrust. These values equate to 105.7% (fully correlated), 100.2% (delta correlated) and 102.3% (3x3 case).

However we believe it is also valuable to show the wind-specific EFL value against wind speed, so that potential trends etc. could be identified.

The conclusion section was edited, hopefully this is more clear now.

Modifications to the manuscript:

Firstly, Section 2.3 was modified to make this point more clear.

Secondly, the conclusion section now entails lifetime fatigue values.

Thirdly, the conclusion section also discusses the quality of the results hopefully more clearly now.

*Comment from Referee 2:*
*EFL — the 1Hz equivalent load. Because Equivalent load can be defined for different frequencies.*

Authors response:
While we fully agree that different sampling frequencies will have a significant impact on the fatigue load results, we have to admit we were not aware of this terminology.
To clarify: Our load data features a sampling frequency of 20Hz. We could imagine it makes sense to normalize by the number of sample points? In our case we chose T to be the number of seconds of the simulation. While this possibly deviates from common practice, it does not affect the results in a critical manner, since both of the loads that we compare have been calculated in a consistent manner.

*Comment from Referee 2:*
*Fig. 7 — Which data points and why are they shifted?*

Authors response:

Here was a mistake in the manuscript, saying the data has been *vertically* shifted. The data was indeed *HORIZONTALLY* shifted so that Gaussian and non-gaussian data points are separated slightly in x-direction.

Modifications to the manuscript:

This has been corrected in the manuscript.

---

## Referee Report (RR1)

(second revision) Comment on "Wind turbine load dynamics in the context of turbulence intermittency"

Jacob Berg, DTU Wind Energy (Referee) jbej@dtu.dk

Thank for your reply – I have additional comments added in red…

Comment from Referee 1:
Exactly how is this correlation constructed?
How is the associated coherence related to atmospheric turbulence?
Authors response:
The focus in this work is on the temporal statistics of the wind. We were able to generate time series with very specific properties and faced the challenge to assemble wind fields based on these very time series.
In one of the extreme cases we prescribe one and the same time series in every of the wind fields grid points in the rotor plane (fully correlated). In another extreme case (delta correlated) we prescribe completely independent time series in every of these grids points, which results in an uncorrelated or delta-correlated field (delta correlated). In order to provide wind fields in between both of these two extreme scenarios, we designed fields in which sub-regions of the grid in the rotor plane are defined, in which we prescribe fully correlated fields. For example, the grid can be sub-divided into 9 regions of a thee by three grid. In these regions the exact same wind time series will be prescribed.
The resulting correlations and coherence must be understood as a simplification of atmospheric turbulence, since atmospheric turbulence features a varying range of temporal and spatial scales.

I still believe that some a little more explanation is needed in the manuscript (you did not add anything) – for example – I don't find the information that the time series are fully correlated within each 3*3 block and zero correlated with the rest - as you state in your comment here – just add two lines... Also, I respect that the study is theoretical and hence focuses on specific mechanisms – but since the conclusions will be used and referred to in many future studies some lines on its relation to realistic atmospheric turbulence (length and time scales) are recommended in order to add relevance for general readers.

Comment from Referee 1:
P6,l25. Can you say something about the physical time scales in the wind model (here defined with constants of physical units). How does for example the integral timescale compare to characteristic time scales of velocity fluctuations in the atmospheric boundary layer like ~10min.
Authors response:
We believe some of the aspects you are asking for are addressed in section 2.1.2. The time series are correlated for roughly 12 seconds, which in comparison to atmospheric turbulence is very short. An integral time scale was not calculated. In our opinion the reason for this is a lack of low frequent dynamics in the velocity signal. As can be seen in e.g. Fig. 1 & 2: There are not too much low frequent dynamics present in our signals. The lack of those might potentially affect the presented results quantitively, but not qualitatively, as the differences in the presented results stem from the intermittency.
We tried to incorporate lower dynamics the velocity signals with our CTRW approach, but essentially found that it would have spoiled other properties of our time signals. Important is to keep in mind that it is our highest priority to have highly comparable Gaussian vs. non-Gaussian fields. We therefore had to trade-off some wind field properties.

Nice discussion – why is that not in the paper?

P7,l5. 1% error in standard deviations of the wind velocity leads to approx. 1% in fatigus loads .
Can you please elaborate – or add reference. So a linear response? – and for all load channels?
Authors response:
We were aiming to provide some reference for the reader how a 1% change in fatigue load can be understood.
The idea was to give a quantitative estimate on how big an offset in turbulence intensity must be, in order to provoke a 1% difference in fatigue loads. We conducted some test simulations with our set-up and found that a 1% increase in turbulence intensity yields roughly a 1% increase in fatigue loading. This is of course highly dependent on the actual turbine, the load sensor, other wind field etc. This is why we added a footnote to emphasize that this result is simply a rough estimation.

1% increase fatigue loading on all turbine load channels (all moments)? Again – the discussion here is much better than the very short statement in the paper.

Comment from Referee 1:
P7,l13. In principle also in the third moment (zero in gaussian stat).
Authors response:
Both of our wind signals (the Gaussian and the non-Gaussian) feature zero skewness. Strictly speaking this is another simplification and deviation from atmospheric turbulence, however it is the aim of this work to focus on the fourth moment and to have no differences in any of the lower order statistics, including the third moment of the 2P statistics.

Sure, but the third order moment and its 4/5 law in some sense can be regarded as the backbone in K41 turbulence – a note of why this is zero in your simulations might me appropriate.

---

## Author Response (AR2)

Dear Editor and Referees,

Thank you very much for the helpful discussion.

As suggested, we incorporated the discussion of the comments 1 to 4 of referee1 and the first two comments of referee 2. We also edited the mentioned typos in the references.

The marked up document compares the earlier revised version of the manuscriptt (version3) against the now second revision (version4?)

[revised manuscript text omitted]